

# Collagen overlays can inhibit leptin and adiponectin secretion but not lipid accumulation in adipocytes

Sherri L. Christian[1], Nikitha K. Pallegar[1], Robert J. Brown[1] and Alicia M. Viloria-Petit[2]

[1] Department of Biochemistry, Memorial University of Newfoundland, St. John's, Newfoundland and Labrador, Canada
[2] Department of Biomedical Sciences, University of Guelph, Guelph, Ontario, Canada

## ABSTRACT

**Background**. White adipose tissue (WAT) is essential for energy storage as well as being an active endocrine organ. The secretion of adipokines by adipocytes can affect whole body metabolism, appetite, and contribute to overall health. WAT is comprised of lipid-laden mature adipocytes, as well as immune cells, endothelial cells, pre-adipocytes, and adipose-derived stem cells. In addition, the presence of extracellular matrix (ECM) proteins in WAT can actively influence adipocyte differentiation, growth, and function. Type I collagen is an abundant fibrous ECM protein in WAT that is secreted by developing adipocytes. However, the extent and overall effect of Type I collagen on adipokine secretion in mature adipocytes when added exogenously has not been established.

**Methods**. We characterized the effects of Type I collagen overlays prepared using two different buffers on adipocyte physiology and function when added at different times during differentiation. In addition, we compared the effect of collagen overlays when adipocytes were cultured on two different tissue culture plastics that have different adherent capabilities. Triglyceride accumulation was analyzed to measure adipocyte physiology, and leptin and adiponectin secretion was determined to analyze effects on adipokine secretion.

**Results**. We found that collagen overlays, particularly when added during the early differentiation stage, impaired adipokine secretion from mature adipocytes. Collagen prepared using PBS had a greater suppression of leptin than adiponectin while collagen prepared using HANKS buffer suppressed the secretion of both adipokines. The use of CellBind plates further suppressed leptin secretion. Triglyceride accumulation was not substantially impacted with any of the collagen overlays.

**Discussion**. Adipokine secretion can be selectively altered by collagen overlays. Thus, it is feasible to selectively manipulate the secretion of adipokines by adipocytes *in vitro* by altering the composition or timing of collagen overlays. The use of this technique could be applied to studies of adipokine function and secretion *in vitro* as well as having potential therapeutic implications to specifically alter adipocyte functionality *in vivo*.

Corresponding author
Sherri L. Christian,
sherri.christian@mun.ca

## INTRODUCTION

White adipocytes serve a crucial function in storing free fatty acids in the form of triglycerides (TG) for energy use (*Berry et al., 2013*) and to prevent toxic deposition of free fatty acids in ectopic sites (*Bays, Mandarino & Defronzo, 2004*). Mature, lipid-laden adipocytes develop from fate-committed pre-adipocytes through the activation of a series of well-regulated transcription factors culminating in the increased expression of CCAAT/enhancer-binding protein alpha (C/EBP-α) and peroxisome proliferator-activator receptor gamma (PPAR-γ), often termed the "master regulators" of adipogenesis. Increased expression of C/EBP-α and PPAR-γ promote the expression of key proteins essential for lipid and carbohydrate storage proteins, such as glucose transporter 4 (Glut4) and perilipin (Plin1), allowing the formation of large lipid droplets that store TG and cholesteryl esters, which are a primary phenotype of terminally differentiated mature adipocytes (*Berry et al., 2013*).

In addition to its energy storing role, white adipose tissue (WAT) can modulate numerous tissues via secretion of adipokines. Key adipokines include adiponectin, which is inversely associated with obesity and increased with fasting, and leptin, which is elevated in obese individuals and decreased with fasting (*Stern, Rutkowski & Scherer, 2016*). Both adiponectin and leptin are secreted from terminally differentiated adipocytes, and they act on WAT as well as distal tissues. For example, adiponectin enhances insulin sensitivity and induces the expansion of WAT, which prevents the toxic deposition of free fatty acids in other organs (*Yamauchi et al., 2001*; *Yamauchi et al., 2002*; *Berg et al., 2001*; *Xu et al., 2003*), whereas leptin acts through the sympathetic nervous system to induce lypolysis of WAT (*Zeng et al., 2015*). In contrast, both adiponectin and leptin can promote glucose uptake by skeletal muscle (*Tomas et al., 2002*; *Bates et al., 2002*). Moreover, leptin promotes the proliferation of breast cancer cells (*Ray, Nkhata & Cleary, 2007*; *Soma et al., 2008*; *Dubois et al., 2014*) while adiponectin inhibits their proliferation (*Li et al., 2011*). Thus, adipokines secreted by WAT have multiple and varied effects on selected tissues and cells.

The role of the extracellular environment on adipocyte function remains incompletely understood (*Huang & Greenspan, 2012*; *Poulos et al., 2015*). Adipocytes are supported by extracellular matrix (ECM) proteins, where laminin, fibronectin, and collagen types I–VI are the major constituents, with the precise ECM composition differing between species and WAT depots (*Mariman & Wang, 2010*). Adipocytes express and secrete ECM proteins, and collagen synthesis during early adipogenesis may promote adipocyte differentiation. Different densities of ECM proteins can cause adipocytes that are grown on ECM scaffolds to alter the *in vitro* secretion of adipokines such as adiponectin and Monocyte Chemoattractant Protein-1 (MCP-1) (*Li et al., 2010*). In addition, pre-adipocytes can actively remodel the ECM via the secretion of matrix metalloproteinases (*Christiaens et al., 2008*).

Systems to model the ECM-adipocyte interactions using collagen-embedded pre-adipocytes or adipocytes for biological study or as a means to engineer tissue for engraftment have been described; however, they are often complicated, require specialized equipment, or do not model the *in vivo* ECM (*Von Heimburg et al., 2003*; *Stacey et al., 2009*; *Chun & Inoue, 2014*). Here, one of our aims was to establish a technique that is simpler than embedding

adipocytes within a three-dimensional matrix, would maintain the pre-adipocyte/adipocyte interaction with the ECM, and would still allow for co-culture analysis with other cell types in future studies. We used this technique to examine the effect of Type I collagen, a highly abundant protein in adipocyte ECM (*Mariman & Wang, 2010*) that is commercially available, and was previously used with success for adipocyte cultures (*Von Heimburg et al., 2003*; *Stacey et al., 2009*; *Chun & Inoue, 2014*). We found a differential effect of the collagen preparation on TG accumulation, and leptin or adiponectin secretion that was further affected by timing and the type of tissue culture (TC) plastic used. These findings suggest that minor manipulations to the ECM surrounding adipocytes can selectively affect their physiology or function.

## MATERIALS AND METHODS

### Cells and Adipogenesis assay

3T3-L1 cells were obtained from the American Type Culture Collection (Manassas, VA, USA) and confirmed to be free of mycoplasma contamination using the MycoAlert assay (Lonza, Basel, Switzerland). All media and supplements were obtained from Invitrogen Life Technologies (Waltham, MA, USA) unless otherwise indicated. Cells were maintained in high-glucose (25 mM) DMEM supplemented with 10% newborn calf serum, 1% penicillin/streptomycin, and 1% sodium pyruvate (DMEM/NCS). Cells were cultured in TC treated 24-well plates (Falcon cat. no. 353226) or CellBind treated 24-well plates (Corning cat. no. 3337). As shown in Fig. 1 and as previously described (*Smith et al., 2015*), pre-adipocytes were plated at $5 \times 10^4$ cells per well to ensure 100% confluency after 24 h and allowed to undergo cell-contact dependent growth arrest for further 24 h. At this time, media were replaced with growth media (high-glucose DMEM supplemented with 10% fetal bovine serum (FBS), 1% penicillin/streptomycin and 1% sodium pyruvate (DMEM/FBS)) containing 0.5 mM 3-isobutyl-1-methylxanthine (IBMX) and 1 µM dexamethasone (Dex) (Millipore, Billerica, MA, USA), and cultured for 48 h. Then the medium was replaced with DMEM/FBS containing 10 µg/ml insulin (Sigma-Aldrich, St. Louis, MO, USA) and cultured for an additional 48 h. Media were then changed to DMEM/FBS and cells cultured for up to five additional days with media replacement every 48 h with DMEM/FBS.

### Collagen matrices

FibriCol® collagen (cat. no. 5133-A) was obtained from Advanced BioMatrix Inc (Carlsbad, CA, USA). The collagen was diluted to 3.3 mg/mL in PBS (1.9 mM $NaH_2PO_4$, 8.4 mM $Na_2HPO_4$, 137 mM NaCl, pH 7.2) or in $1\times$ HANKS buffer (1.26 mM $CaCl_2$, 0.81 mM $MgSO_4$, 5.4 mM KCl, 0.44 mM $KH_2PO_4$, 137 mM NaCl, 0.34 mM $Na_2HPO_4$, 5.5 mM D-glucose, 0.05 mM Phenol Red sodium salt, obtained as $10\times$ concentrate cat. no. 0919101-54, MP Biomedicals, Santa Ana, CA, USA). Collagen in PBS (PBS-collagen) was neutralized to pH 7–7.5 by addition of HCl prior to layering onto cells. Collagen in HANKS was neutralized to pH 7–7.5 by addition of 250 mM HEPES (pH 7.2) (HANKS-collagen). Collagen solutions or control solutions lacking collagen (300 µL) were gently layered on top of the adipocytes and allowed to polymerize at 37 °C for 40 min, thus generating a collagen layer of an approximate thickness of 1.5 mm. For adipocytes not treated with collagen or

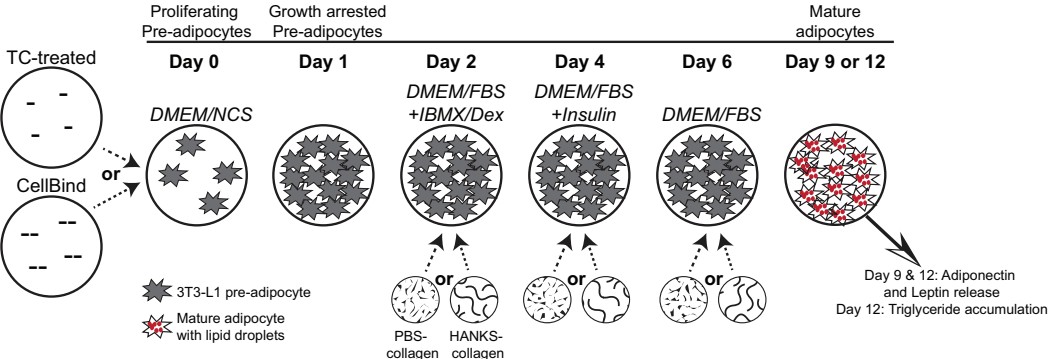

**Figure 1  Schematic diagram of 3T3-L1 adipogenesis and timing of collagen overlays.** Pre-adipocytes were cultured for 24 h on regular tissue-culture (TC) treated or CellBind plates, until they underwent contact-dependent growth arrest. Collagen, or buffer only control, was overlaid at days 2, 4, or 6 and allowed to solidify for 40 min at 37 °C. Medium was overlaid onto the collagen or buffer after the 40 min polymerization period. DMEM/FBS medium containing IBMX and dexamethasone (DMEM/FBS+IBMX/Dex) was added at day 2, DMEM/FBS with insulin (DMEM/FBS+Insulin) was added at day 4, and DMEM/FBS was added at day 6 and replaced every 48 h. Pre-adipocytes, prior to induction of differentiation, are indicated by starburst shapes. Lipid droplets in mature adipocytes are shown with small red circles. Addition of PBS-collagen or HANKS-collagen is indicated at day 2, 4, or 6. Adiponectin and leptin concentrations were determined at day 9 and day 12, and TG accumulation was determined at day 12.

buffer controls at that particular stage, medium was not removed. At the end of the 40 min, after the collagen solutions were polymerized, 1 mL of medium, appropriate for the stage of differentiation, was layered on top of the collagen layer or buffer solution (Fig. 1). At day 2, the medium overlaid was DMEM/FBS with IBMX/Dex (DMEM/FBS+IBMX/Dex). At day 4, the medium overlaid was DMEM/FBS with insulin (DMEM/FBS +Insulin). At day 6, the medium overlaid was DMEM/FBS and replaced with DMEM/FBS every 48 h.

## Triglyceride quantification

Media were removed at day 12 (Fig. 1); cells and collagen was washed twice with PBS and then removed by scraping followed by re-suspension in 1 mL PBS. Lipids from the re-suspended cells were extracted using the Bligh–Dyer method (*Bligh & Dyer, 1959*), followed by additional manipulations as follows. Following the removal of the organic solvent layer from the Bligh-Dyer extraction, lipids remaining in the upper phase were re-extracted (in duplicate) by adding 2.5 mL of chloroform, vortexing for 30 s, and centrifuging at $1,000 \times g$ for 5 min. The organic solvent layer was removed and pooled with the organic solvent layer from the Bligh-Dyer extraction. The pooled extracts were back extracted by mixing with an equal volume of PBS, vortexing for 30 s, and centrifuging at 1,000 g for 5 min. The organic solvent layer was removed, dried under $N_{2(g)}$, re-suspended in 500 μL isopropanol, and stored under $N_{2(g)}$ at −20 °C until needed. To control for extraction efficiency, 10 μg of a TG standard (cat. no. 17810; Sigma Aldrich, St. Louis, MO, USA) was extracted as above, and the TG quantified with 25 μL of extracted sample ($n = 11$) was compared to the TG quantified from 25 μL of 0.02 μg/μL TG standard (in isopropanol). TG cellular accumulation was corrected for extraction efficiencies within each extraction procedure

(mean extraction efficiency was 88.2% with a range of 79.8%–97.7%). A colorimetric commercial kit from Wako Diagnostics (Richmond, VA, USA) was used to quantify TG, using a standard curve of 0–50 $\mu$g TG.

### Adiponectin and leptin quantification

Mouse leptin and adiponectin concentrations in the culture supernatant were determined using DuoSet ELISA kits specific for mouse from R&D Systems (Minneapolis, MN, USA) following the manufacturer's instructions. Samples were analyzed in duplicate. A 4-parameter log–logistic model was used to fit the data to the standard curves run simultaneously using R v3.0 (*R Core Team, 2015*). Calibration curves were run to ensure accurate dilution of the supernatants and found to require dilution of 1,024-fold for adiponectin and no dilution for leptin. Samples where all replicates from one treatment group had an absorbance below background are indicated as not detected (ND). For samples with levels of leptin or adiponectin below the level of detection, the amount was set to 8 pg/mL for leptin and 0.035 ng/mL for adiponection, which was just below the lowest detectable concentration of 8.9 pg/mL and 0.039 ng/mL, respectively. Analysis of differences were performed using Wilcoxon rank-sum analysis as samples with undetectable levels can be included without compromising the analysis.

### Statistical analysis

Statistical analysis was performed in R v3.0 (*R Core Team, 2015*), as indicated in the respective figure legends. Experiments were repeated at least three times beginning with a unique passage of 3T3-L1 cells. Each independent experiment was considered a biological replicate. Differences were considered significant at $P < 0.05$.

## RESULTS

### Collagen matrices influence adipocyte function

We found that there was a statistically significant decrease in overall TG accumulation with collagen overlays when assessed at day 12 (Fig. 2A). However, when we analyzed the effects of the overlay at each timepoint, we found that there was no significant effect of PBS-collagen on the total TG accumulation at any specific stage. In contrast, addition of HANKS-collagen on day 2, but not day 4 or 6, resulted in a statistically significant reduction in total TG (Fig. 2A).

We then analyzed leptin secretion at days 9 (Fig. 2B) and 12 (Fig. 2C), and we compared the amount of leptin detected in the supernatant when collagen overlays were added at day 2, 4, or 6. Overall, we found that the addition of either PBS-collagen or HANKS-collagen had an overall effect of suppressing leptin accumulation when analyzed at both days 9 and 12. The suppression was greater when the collagen was added at day 2 or day 4 compared to when added at day 6, with no leptin detected after collagen addition in most cases (Figs. 2B–2C). Addition of collagen at day 6 did not significantly affect the amount of leptin detected at day 9 but suppressed the amount of leptin detected at day 12 with HANKS-collagen only. At day 9, the overall effect of PBS-collagen was statistically different from HANKS-collagen, with statistically similar effects at day 12.

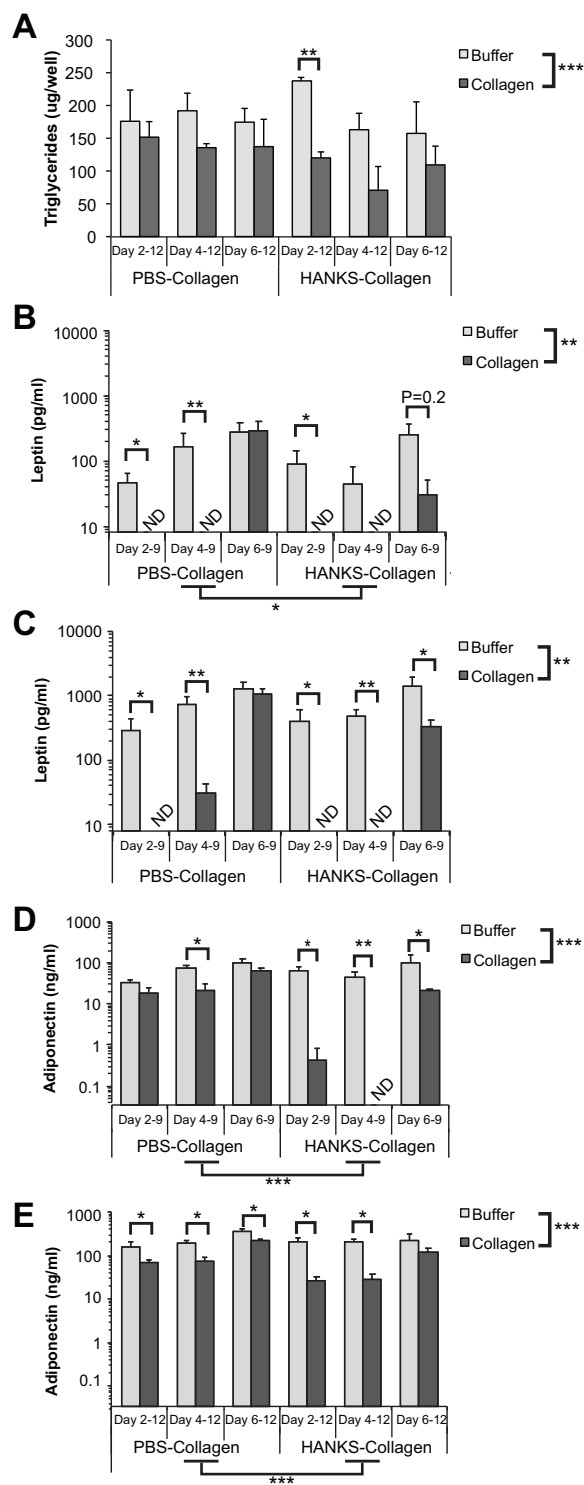

**Figure 2** **Triglyceride (TG), leptin, and adiponectin production are differentially affected by PBS-collagen and HANKS-collagen when cells are plated on regular TC-treated plates.** (continued on next page...)

**Figure 2 (…continued)**
(A) Total TG in 3T3-L1 cells at day 12 after adding collagen overlays at days 2, 4 or 6. (B–E) Levels of secreted leptin and adiponectin were determined at day 9 (B, D) and day 12 (C, E). Note that leptin and adiponectin concentrations are shown in Log10 scale as the changes in concentration can span multiple orders of magnitude. Statistical analysis was done by 3-way ANOVA. *A priori* analysis of control vs. collagen at each stage by *T*-test for TG and adiponectin and by Wilcoxon rank sum for leptin levels as some samples were below the detection limit of the ELISA (ND, not detected). *$P < 0.05$, **$P < 0.01$, ***$P < 0.001$, shown are mean $\pm$ sem of 3 independent biological replicates.

Adiponectin secretion was affected more by HANKS-collagen at both day 9 (Fig. 2D) and 12 (Fig. 2E) when compared to PBS-collagen. Addition of HANKS-collagen, but not PBS-collagen at day 2, resulted in a large and significant inhibition of adiponectin secretion at day 9 (Fig. 2D). By day 12, adiponectin levels remained significantly lower in adipocytes with HANKS-collagen added at day 2 or 4, but not at day 6, suggesting the effect is time- or maturation-dependent (Fig. 2E). Addition of PBS-collagen significantly, but modestly, suppressed adiponectin secretion by day 12 and at day 9 when added at day 4 only.

## Effect of CellBind plates on adipocyte physiology and function in the presence of collagen

The previous experiments were performed using typical TC treated plastic. However, culturing 3T3-L1 cells on specially treated dishes with higher levels of incorporated oxygen on the plastic, branded CellBind (*Pardo et al., 2010*), can increase the ease of the adipogenesis assay because the cells remain more firmly attached. This effect is noticed particularly after IBMX and Dex treatment at day 2 when the cells have a more rounded morphology (SL Christian & NK Pallegar, 2015, unpublished data). Therefore, we sought to determine if the addition of collagen overlays in combination with CellBind plates would affect adipogenesis.

We first determined if CellBind plates affected adipocyte physiology or function in the absence of collagen. As expected, we found a significant increase in the amount of cellular TG, as well as a significant increase of secreted leptin and adiponectin, when cells were induced to undergo adipogenesis compared to control (Neg) cells (Figs. 3A–3C). We observed a trend towards an increase in TG accumulation when adipogenesis was induced in the CellBind plates compared to regular TC plates ($P = 0.07$). Overall, there was no significant effect of plate type on leptin or adiponectin secretion. However, we observed spontaneous release of adiponectin on CellBind plates from control cells (Neg) resulting in no significant increase in adiponectin secretion detected at day 9 in response to adipogenesis induction (Pos) (Fig. 3C).

We next determined the effect of the collagen overlays on TG accumulation and adipokine secretion when CellBind plates were used. Similar to the regular TC plates, we found an overall suppression of TG accumulation in the presence of the collagen overlays (Fig. 4A). However, the only significantly different pairwise comparison was with PBS-collagen added at day 4. There was no overall significant difference between PBS-collagen and HANKS-collagen on TG accumulation.

In contrast, we found that leptin levels on day 9 were particularly disrupted when cells were treated with either buffer alone or collagen in either type of buffer during

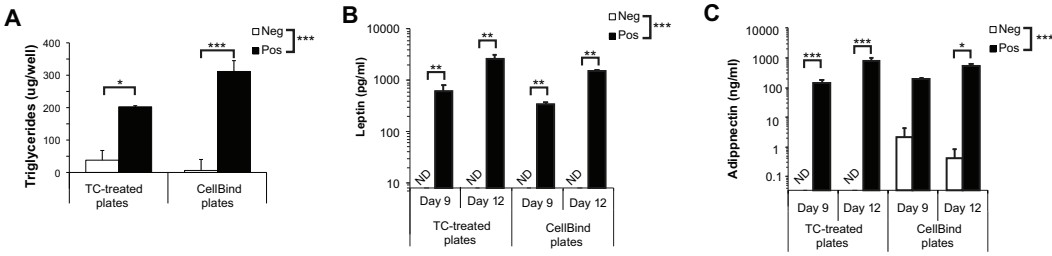

**Figure 3** **Adipogenesis is not significantly different when 3T3-L1 cells are cultured in regular tissue-culture (TC) treated dishes or CellBind dishes.** (A) Triglyceride accumulation at day 12, (B) leptin and (C) adiponectin secretion at days 9 and 12 in cells culture on regular TC or CellBind plates in the absence (Neg) or presence (Pos) of adipogenic inducers as described in the methods, with no buffer incubation steps. Note that leptin and adiponectin concentrations are shown in Log10 scale as the changes in concentration can span multiple orders of magnitude. Statistical analysis by 2-way ANOVA. *A priori* analysis of control vs. collagen at each stage by *T*-test for TG and adiponectin and by Wilcoxon rank sum for leptin levels as some samples were below the detection limit of the ELISA. *$P < 0.05$, **$P < 0.01$, ***$P < 0.001$. Mean ± sem shown of 3–4 independent biological replicates.

adipogenesis (Fig. 4B), suggesting that leptin secretion from cells grown on CellBind plates is more sensitive to any environmental disruption during adipogenesis, which occurs in both conditions. However, by day 12 the amount of leptin detected from the cells without collagen was similar to cells grown without disruption (compare Fig. 4C to Fig. 3B). Addition of HANKS-collagen at day 4, but not day 2 or 6, resulted in a significant decrease in leptin accumulation at day 12 (Fig. 4C). In addition, there was a significant overall difference between the addition of PBS-collagen compared to HANKS-collagen in leptin accumulation at day 12, with HANKS-collagen having a greater impact.

There was an overall significant decrease in adiponectin accumulation at days 9 and 12 with the addition of collagen (Figs. 4D–4E). Overlaying HANKS-collagen at day 4 resulted in a significant decrease in adiponectin detected at both day 9 and day 12. Overlaying PBS-collagen at day 4 caused a significant decrease when detected at day 12 but not day 9. Moreover, the magnitude of the decrease in adiponectin accumulation was much less than with TC plates (compare Figs. 4D–4E to Figs. 2D–2E).

## DISCUSSION

Our goal was to determine if collagen matrix overlays would impact adipocyte physiology, as measured by TG accumulation, and function, as measured by adipokine secretion. We found that both physiology and function could be differentially affected by the addition of collagen overlays in a manner that depended on when during adipogenesis the collagen was added. Moreover, collagen overlays prepared using different buffers had differential effects on adiponectin and leptin secretion.

Overall TG accumulation was generally suppressed by the addition of both PBS-collagen and HANKS-collagen (Figs. 2A and 4A). However, only two timepoints showed a specific reduction in TG accumulation when compared to without buffer alone, with neither of the changes particularly substantial. Therefore, we conclude that the collagen matrix

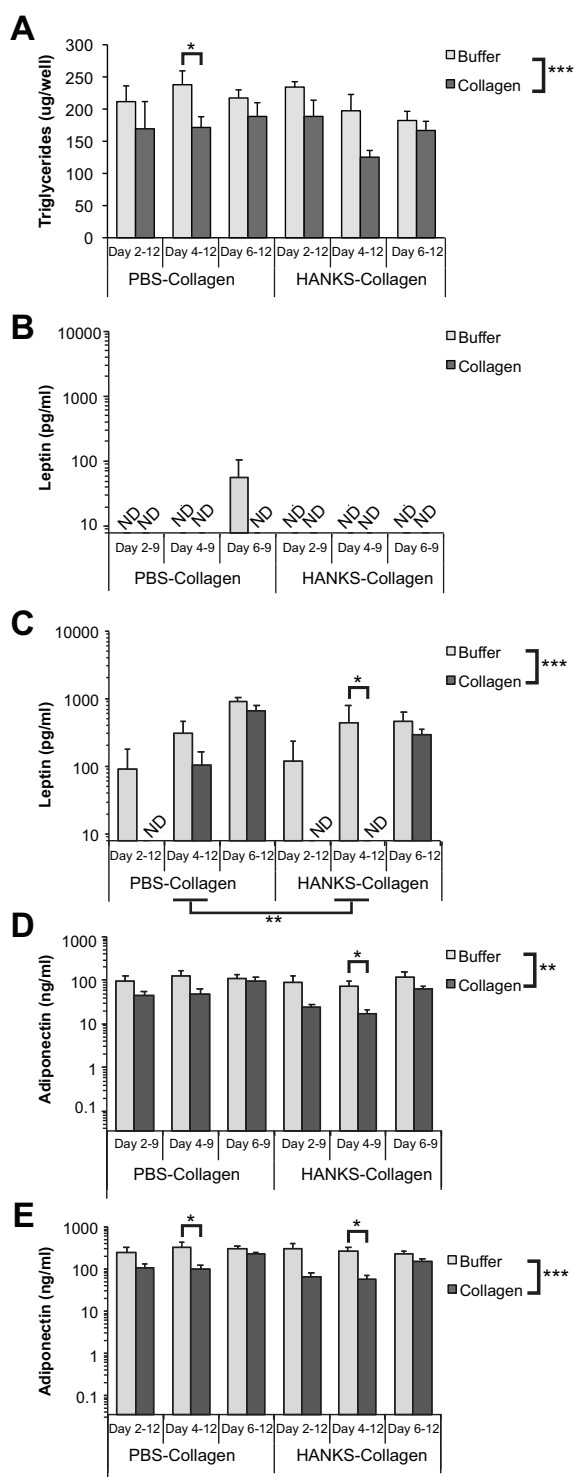

**Figure 4** Triglyceride (TG), leptin, and adiponectin production are differentially affected by PBS-collagen and HANKS-collagen when cells are plated on CellBind plate. (continued on next page...)

**Figure 4 (…continued)**
(A) Total TG in 3T3-L1 cells at day 12 after adding collagen overlays at days 2, 4 or 6. (B–E) Levels of se-creted leptin and adiponectin were determined at day 9 (B, D) and day 12 (C, E). Note that leptin and adiponectin concentrations are shown in Log10 scale as the changes in concentration can span multiple orders of magnitude. Statistical analysis by 3-way ANOVA. *A priori* analysis of control vs. collagen at each stage by *T*-test for TG and adiponectin and by Wilcoxon rank sum for leptin levels as some samples were below the detection limit of the ELISA (ND, not detected). *$P < 0.05$, **$P < 0.01$, ***$P < 0.001$. Mean $\pm$ sem of 3 independent biological replicates is shown.

overlays do not substantially alter the normal accumulation of lipid in mature adipocytes. It remains to be determined if the modest decreases in TG accumulation would translate to biologically relevant outcomes in an *in vivo* situation.

Our analysis of adipocyte function as determined by adipokine secretion revealed that leptin and adiponectin can be affected by both the method of collagen matrix preparation and the type of TC plastic used. We do not believe that the reduction in adipokine detection is due to collagen physically blocking the passage of secreted proteins that were subsequently detected by ELISA. Leptin has a calculated mass of approximately 19 kDa protein while adiponectin has a calculated mass of 26 kDa. Since leptin was impacted more than adiponectin, it is unlikely that the collagen matrices are reducing the passage of secreted proteins based on size. Adiponectin is glycosylated in at least 6 sites (*Richards et al., 2006*; *Richards et al., 2010*), whereas post-translational modifications of leptin, with the exception of disulfide bonds, have not been reported. The predicted pI of leptin is 5.85, while the predicted pI of unmodified adiponectin is 5.57, suggesting that different charges on the unmodified proteins of $-2.5$ and $-6.4$, respectively, could potentiate charge-charge interactions between the adipokine and collagen. In fact, an interaction between adiponectin and collagen has been reported previously *in vitro* and in injured but not healthy blood vessels *in vivo* (*Okamoto et al., 2000*). However, leptin has not been reported to interact with collagen. Moreover, equivalent amounts of leptin and adiponectin were secreted when the collagen was added to more mature cells (i.e., at day 6). Thus, there is no evidence to suggest that the collagen matrices physically impede the release of adipokines into the surrounding media.

The adipocytes were exposed to an absence or reduction of glucose during the 40 min collagen polymerization period for PBS and HANKS-based buffers, respectively. Reduced adipogenesis is observed when cells are exposed to low glucose levels during early phases of differentiation, corresponding to days 2–5 in our study, but not at later stages (*Jackson et al., 2017*). The levels of TG, adiponectin, and leptin secreted by cells exposed to no/low glucose buffer were similar to the controls by day 12 but generally lower at day 9 (compare Fig. 2 to Figs. 3 and 4). Therefore, it appears that the cells are able to overcome any effects of the short-term low glucose exposure given enough time for full differentiation and maturation. However, differentiation of cells in the CellBind plate when combined with the 40 min buffer or collagen incubation further reduced leptin secretion, particularly at day 9, suggesting that firm binding of the cells to the plates affects their ability to withstand other stressors, such as reduced glucose levels. Therefore, it is possible that the combination

of reduced glucose in the presence of monomeric or polymerized collagen is necessary for the full reduction in adipocyte function.

It is possible that, even after addition of media with sufficient glucose levels, the collagen layers could impact nutrient or oxygen availability to the cells due to physical interference of advective mixing. Unfortunately, it was not possible to control for this as the buffer solutions, used as controls for the absence of media during polymerization, do not form discrete layers. In fact, the buffer dilutes the media by ∼30% thus potentially decreasing the nutrient availability as well as the adipokine concentration compared to the wells with polymerized collagen. Therefore, we may be underestimating the reduction in adipokine or TG secretion. Future studies will be required to determine the concentration of key nutrients and oxygen in the collagen layer or at the collagen-adipocyte interface in comparison to the buffer controls to determine if this could be causing the effects we observed.

The collagen overlays could also decrease the clonal expansion that occurs during adipogenesis of 3T3-L1 cells (*Gregoire, Smas & Sul, 1998*; *Tang, Otto & Lane, 2003*). This re-entry into the cell cycle after contact-dependent growth arrest occurs during the early phases of adipogenesis and could therefore be impacted by addition of collagen at day 2 or 4. If this were the case, we could expect a significant reduction in TG accumulation as, in theory, there would be a reduction of up to 50% of the cells. Since we have not observed a significant or substantial reduction in TG accumulation for most of the pairwise analysis, we do not believe that this is the major mechanism. However, precise analysis of DNA replication will be required to conclusively establish if the collagen overlays affect clonal expansion.

Leptin was strongly affected by the addition of both types of collagen matrices. Strikingly, addition of collagen at the earlier stage of adipogenesis (day 2 or day 4) resulted in leptin secretion that was, in many cases, below the level of detection. Thus, addition of collagen during active differentiation decreases leptin secretion in a manner that is not readily restored by additional time in culture. Leptin expression is regulated by a variety of transcription factors including C/EBPα, SREBP1, and FosL1 during adipogenesis (*Miller et al., 1996*; *Mason et al., 1998*; *Wrann & Rosen, 2012*). In addition, leptin expression is promoted by insulin, glucocorticoids, and even leptin itself (*Wrann & Rosen, 2012*). The ability of collagen or buffer to impair leptin secretion when added at early stages of adipogenesis but not late stages suggests that these manipulations may be interfering with the action of a regulator that acts early during adipogenesis. Elucidating the mechanism for this stage-dependent effect will be an important focus for future study.

In contrast to leptin, effects of collagen overlays on adiponectin secretion were modest. PBS-collagen had very little impact on adiponectin secretion while HANKS-collagen, especially when associated with TC-treated plates measured at day 9, substantially decreased the secretion of adiponectin. Adiponectin expression is regulated by insulin via activation of PPAR- γ as well as by other transcription factors such as C/EBPα and negatively regulated by FoxO1 (*Shehzad et al., 2012*). Thus, it appears that addition of HANKS-collagen significantly impairs the regulation of adiponectin synthesis and/or secretion, but this effect is reduced when cells are cultured in CellBind plates. While it is

clear that collagen overlays differentially regulate the appearance of adiponectin and leptin, the precise mechanism for this regulation will require further study.

It is not clear why the HANKS-collagen had different effects than PBS-collagen on adipokine secretion when compared to their respective buffer controls. While we observed that the HANKS-collagen solidified at a slower rate than PBS-collagen, both were solidified completely within 24 h. The HANKS buffer contains potassium ions, $CaCl_2$, $MgSO_4$, and D-glucose, which are absent in the PBS-collagen preparation. The differences in buffer compositions did not cause obvious changes to the collagen overlays at the macroscale (data not shown). We have not found any published studies on the specific interactions between these molecules and collagen that may explain the differing cellular responses. Therefore, future study will be necessary to dissect out the key factors that cause the functional changes to the collagen matrices.

Regardless of the mechanism, we have clearly shown that manipulations to the preparation of Type I collagen can result in significant and specific alterations to adipocyte physiology and function. Thus, selective alteration of adipokine secretion without impacting TG storage may be possible using collagen matrices. Adapting this method to *in vivo* procedures could potentially be used to modify appetite-regulating hormones without diminishing the essential role of adipocytes to store free fatty acids, which are toxic in high abundance or when deposited ectopically (*Bays, Mandarino & Defronzo, 2004*).

## CONCLUSIONS

In summary, we found that the secretion of leptin and adiponectin can be selectively manipulated while not substantially impairing TG synthesis in developing adipocytes by use of different collagen preparations and cell culture plates. These findings may provide researchers a new way to affect adjacent cells or tissue by selectively manipulating adipocyte function *in vitro* or, potentially *in vivo*.

## ACKNOWLEDGEMENTS

We thank Zhe Dong and Erika Merschrod for helpful discussion and for providing the collagen.

### Funding

This work was funded by Canadian Institutes of Health Research operating grants no. 126614 and 126754, with matching funds from the Research and Development Corporation of Newfoundland grant 5404.1090.103, to Sherri L. Christian, and by a Natural Sciences and Engineering Research Council of Canada Discovery grant (402185-2011-RGPIN) to Robert J. Brown. Nikitha K. Pallegar is supported by a trainee award from the Beatrice Hunter Cancer Research Institute with funds provided by the Terry Fox Strategic Health Research Training Program in Cancer Research at CIHR and by Memorial University of

Newfoundland. The funders had no role in study design, data collection and analysis, decision to publish, or preparation of the manuscript.

## Grant Disclosures
The following grant information was disclosed by the authors:
Canadian Institutes of Health Research: 126614, 126754.
Research and Development Corporation of Newfoundland: 5404.1090.103.
Natural Sciences and Engineering Research Council of Canada Discovery: 402185-2011-RGPIN.
Beatrice Hunter Cancer Research Institute.
The Terry Fox Strategic Health Research Training Program in Cancer Research.

## Competing Interests
The authors declare there are no competing interests.

## Author Contributions
- Sherri L. Christian conceived and designed the experiments, performed the experiments, analyzed the data, contributed reagents/materials/analysis tools, prepared figures and/or tables, authored or reviewed drafts of the paper, approved the final draft.
- Nikitha K. Pallegar performed the experiments, analyzed the data, authored or reviewed drafts of the paper, approved the final draft.
- Robert J. Brown conceived and designed the experiments, performed the experiments, analyzed the data, authored or reviewed drafts of the paper, approved the final draft.
- Alicia M. Viloria-Petit conceived and designed the experiments, contributed reagents/materials/analysis tools, authored or reviewed drafts of the paper, approved the final draft.

## Data Availability
The raw data are provided in Data S1.

## Supplemental Information
Supplemental information for this article can be found online at http://dx.doi.org/10.7717/peerj.4641#supplemental-information.

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
