# Peer review of "Collagen overlays can inhibit leptin and adiponectin secretion but not lipid accumulation in adipocytes"

_PeerJ, doi:10.7717/peerj.4641_

## Round 0.1 · original submission · Minor Revisions

As you will see, the tone of all three reviewers is largely positive. Reviewer 2 has concerns regarding the glucose concentrations chosen - though this may reflect ambiguity in your methodological description. This point needs careful attention. I would also like to see consideration given to the points from Reviewer 1 on the ELISA assays. These would appear the most substantive points that require detailed and careful attention. The other points should all be fairly easy to address.

Reviewer 1 ·

Basic reporting

The authors investigate the effect of collagen on the secretion of leptin and adiponectin in developed adipocytes. The paper is well written, succinct and objective. Structure appears appropriate. Raw data is included in supplemental file.

Intro provides context, with appropriately referenced literature. Authors should consider the paper by Q Li et al titled “The density of extracellular matrix proteins regulates inflammation and insulin signalling in adipocytes” (FEBS Letters 2010) that addresses a similar question, although in 2D, regarding collagen coating density on adiponectin secretion.

Figures are relevant, but some labels could be improved for clarity. For instance, what is the meaning of “Ins2d” “+I/D” and “+Ins” in Fig 1?

ELISA measurements are presented on a log-scale. This is likely done because in some cases, the differences do indeed span several orders of magnitude. However, the lower limit of detection is 62.5 pg/ml for the leptin (line 132; does the same limit apply for adiponectin?). Some of the plotted values may be near this cut-off (e.g., some values in Fig 2 B&C), and it’s difficult to tell on a log-scale how close these values are to the cut-off. As there is no reason to show values below the lower cut-off, the authors should consider plotting the data with the lower cut-off as the bottom.

Experimental design

This original primary research paper fits within the scope of the PeerJ journal. This research question is well defined and meaningful. The study aims to fill a gap in knowledge regarding how the 3D ECM environment affects adipocyte function. The gap exists because most studies of adipocytes use 2D surfaces.

Approximately 300 ul of collagen solution is added to the adipocytes (line 106). How thick of a layer does this produce (a rough estimate in my hands gives around 1.6 mm)? The motivation starting on line 76 gives the merits of a 3D environment, but the cells experience a rigid 2D surface surrounded by softer collagen. Did the authors visualise the cells to see if they migrated into the 3D collagen, or whether they remained attached to the plate?

Would the increased thickness of collagen, because it reduces advective mixing within the well, reduce oxygen or nutrient transport and thereby affect adipocyte function independent of collagen per se?

This reviewer has two questions regarding the ELISA measurements. First, the authors provide reasonable arguments against a deleterious effect of collagen on the ELISA measurements (lines 217-233). However, some of this concern (particularly regarding binding) could in principle be abated by performing a calibration curve in the presence of the collagen layer. Further, by the collagen occupying some physical volume (albeit small), there may be a steric hindrance effect that may influence the measured concentration. For these reasons, it would be useful to show a calibration curve both with and without the collagen. Second, what is the rationale for setting below detection limit ELISA values to 2 pg/ml, when presumably values up to 62.5 pg/ml would yield an undetectable result? Perhaps with a rank-sum analysis, this doesn’t make much difference in the statistics, but this seems arbitrary. Can this be clarified a bit more please?

Please state whether cells treated with PBS are deprived of glucose. Lines 102-103 do not mention glucose, which is typically absent from PBS, while the Hank’s buffer contains glucose (line 258; but at what concentration?). As the cells are cultured in high glucose medium (lines 91 and 94), glucose removal may therefore occur as a shock. Although I acknowledge that both PBS and PBS+Collagen groups would be similarly deprived, the absence of glucose seems peculiar. Lines 108-109 suggest that proper DMEM/FBS was returned at the “day indicated (Fig. 1)”. Perhaps this addresses the concern above, but if so, this is not entirely clear, in the text or in Figure 1.

It may have been useful to measure cell numbers. Cells in 3D environments tend not to divide as rapidly as those in 2D environments, and thus there could possibly have been a difference in cell numbers between buffer and collagen-treated samples, that could have contributed to the measured differences in TG or ELISA results.

Validity of the findings

Overall, the results appear reasonable sound subject to the questions or ambiguities raised above.

Is there an inconsistency between results shown in Figs 2B, 3B and 4B? In Fig 3B, cells cultured directly on either plate show strong leptin expression by 9 and 12 days. In Fig 2B, cells cultured directly on TC plates without collagen also show reasonable (but lower, relative to Fig 3B) leptin expression even with buffer disruption, while on the CellBind plate the leptin response is virtually eliminated. The explanation in the Discussion points to the disruption caused by buffer washes, but if this was the explanation, then why is response observed in Fig 2B and not Fig 4B?

Reviewer 2 ·

Basic reporting

The manuscript is clearly written using a professional English language style.
The Introduction is well referenced and the context providing a rationale for the work to be undertaken and the references used to support the work are appropriate.

The figures produced are relevant and appropriate but there is ambiguity in understanding and interpreting the results due to a poor experimental design. More detail in Figure 1 outlining the design would be helpful and provide a more accurate reflection of the work carried out.

Experimental design

This is an area of real interest and raises a cautionary note when comparing results between different groups who may be using different methods. However, the research question is not well defined and the experimental design reflects this. The manuscript seeks to determine the effect of collagen overlay on the expression of leptin and adiponectin and triglycerides in 3T3 L1 adipocytes.

My comments below reflect my areas of concern and consequent ambiguity of findings:

1) Apparent poor use of controls to reflect glucose concentration:
a) Line 91: Cells maintained in high glucose DMEM – this is equivalent to 25mM glucose. The collagen was prepared in 1X HANKS – I have been unable to access the precise details from the product number provided but standard HANKS would be 5.5 mM. There is nothing in the manuscript that alludes to the rationale for undertaking this with high glucose and given previous work demonstrating that glucose controls adipogenesis in 3T3 cells (Jackson RM, Griesel BA, Gurley JM, Szweda LI, Olson AL Glucose availability controls adipogenesis in mouse 3T3-L1 adipocytes via up-regulation of nicotinamide metabolism. J Biol Chem. 2017 Nov 10;292(45):18556-18564. doi: 10.1074/jbc.M117.791970. Epub 2017 Sep 15.) it would seem appropriate to carefully control the concentration of glucose of each replicate. While this may have been carried out, the description does not reflect it. Line 101-109 outlines the preparation of the collagen to be in either HANKS or PBS. Again there is a clear difference in the glucose concentration between the PBS and the HANKS and yet there appears to be no mention of this at any point during the manuscript.
b) The samples that did not contain any collagen presumably were either the corresponding PBS or HANKS, did not polymerize (line 107). It is not clear if the control solutions were added to cells containing media but if so this would have acted to dilute all the constituents in the media. In addition the authors state that the additives appropriate to differentiation stage was added (line 109) as illustrated in Fig 1. One presumes this is the +I/D, +lns and Lns2d. However, the details of the meaning of the symbols was not evident in the figure legend. Greater clarity in the presentation of the data may clarify current concerns regarding controls and glucose concentration.
c) Lack of acknowledgement of gaseous perfusion: going hand in hand with the glucose concentration difference and the cells that have a collagen overlay will have less access to oxygen. This will affect glucose metabolism and resulting changes may at least in part be reflected by this difference. A concentration of 3.3% is still likely to allow for diffusion of the oxygen but it will be delayed when compared to those in the absence of an overlay and I wonder if the authors have determined the oxygen levels in the overlay? This should be acknowledged in the interpretation of any results observed.
2) Lack of documented morphology of cells in different treatment groups: the authors refer to the rounded nature of the cells on the CellBind plate (lines 168-170) and wonder if any of the other treatments caused morphological changes and if so, this may be worthwhile to include in the results rather than to refer to them as unpublished observations.

Validity of the findings

The manuscript reports some interesting findings but with cautionary note based on previous comments.
Results show that the timing of the addition of the matrix will affect production of TG, leptin and adiponectin. The author’s comment that this is likely to be due to different stage of differentiation is supported by the data and the general increase in levels over time also supports this. The effect on adiponectin were reported to be more dependent on collagen preparation “line 245-248. This is possible, but the contribution from differences in glucose concentration, oxygen content and differentiation status is difficult to interpret without further details and will be a consequence of comparing PBS with HANKS. Furthermore, the measurement of some of the down-stream regulators of these proteins would consolidate these conclusions.

Additional comments

The manuscript highlights some important aspects of cell culture with the environment having a profound effect on TG, AD and Leptin. This will clearly affect the cells and caution should be applied to articles where the response of cells is varied and where a different culture model is used. Greater clarity of the research design would improve ability to interpret this data for example images of cells at different stages, more detailed figure 1 including acronyms used and comparison with controls. To combine this with an extended section for methods on a) cells and reagents and b) collagen matrix to include how the collagen was used and applied to the cells to facilitate understanding of context.

Reviewer 3 ·

Basic reporting

1. BASIC REPORTING
In general this is a well written manuscript, with an appropriate introduction and discussion. There is a good literature review.

Experimental design

2. EXPERIMENTAL DESIGN
The article is in scope and looks rigorous. Good detailed methods. The authors used three-way ANOVA in Figs 2 & 4 but two-way ANOVA was sufficient in Fig 3 for (what looks like) datasets with a similar complexity. More justification for the use of the different statistical methods would be desirable in the Methods.

Validity of the findings

3. VALIDITY OF THE FINDINGS
In general, the data is robust and the discussion sound

Additional comments

4. GENERAL COMMENTS
None.

---

## Round 0.2 · accepted · Accept

Your detailed response to each of the points raised was appreciated.

#